# Intestinal Microbiome Profiles in Broiler Chickens Raised with Different Probiotic Strains

**DOI:** 10.3390/microorganisms12081639

**Published:** 2024-08-10

**Authors:** Julia Marixara Sousa da Silva, Ana Maria De Souza Almeida, Ana Carolina Borsanelli, Flávia Regina Florencio de Athayde, Eduardo de Paula Nascente, João Marcos Monteiro Batista, Alison Batista Vieira Silva Gouveia, José Henrique Stringhini, Nadja Susana Mogyca Leandro, Marcos Barcellos Café

**Affiliations:** 1Veterinary and Animal Science School, Federal University of Goiás, Goiania 74605-080, Goiás, Brazil; marixaraj@gmail.com (J.M.S.d.S.); ana_almeida@ufg.br (A.M.D.S.A.); anaborsanelli@ufg.br (A.C.B.); eduardodepaula100@gmail.com (E.d.P.N.); joao1992marcos@outlook.com (J.M.M.B.); alisonmestre28@gmail.com (A.B.V.S.G.); henrique@ufg.br (J.H.S.); mogyca@ufg.br (N.S.M.L.); 2Faculty of Veterinary Medicine of Araçatuba, São Paulo State University (UNESP), Araçatuba 16050-680, São Paulo, Brazil

**Keywords:** alternatives to antibiotics, poultry farming, microbiome, modulation, intestinal health

## Abstract

The composition of the intestinal microbiota can influence the metabolism and overall functioning of avian organisms. Therefore, the objective of this study was to evaluate the effect of three different probiotics and an antibiotic on the microbiomes of 1.400 male Cobb^®^ broiler raised for 42 days. The experiment was conducted with the following treatments: positive control diet (basal diet + antibiotic); negative control diet (basal diet without antibiotic and without probiotic); basal diet + Normal Avian Gut Flora (NAGF); basal diet + multiple colonizing strain probiotics (MCSPs); and basal diet + non-colonizing single strain probiotics (NCSSPs). The antibiotic (enramycin—antibiotic growth promoter) and probiotics were administered orally during all experiment (1 to 42 days), mixed with broiler feed. To determine the composition of the microbiota, five samples of ileal digesta were collected from 42-day-old chickens of each experimental group. The alpha and beta diversity of the ileal microbiota showed differences between the groups. MCSPs presented greater richness and uniformity compared to the positive control, negative control, and NCSSPs treatments, while the negative control exhibited greater homogeneity among samples than NCSSPs. MCSPs also showed a higher abundance of the genus Enterococcus. There were differences between the groups for low-abundance taxa (<0.5%), with NAGF showing higher levels of Delftia, Brevibacterium, and Bulleidia. In contrast, NCSSPs had a higher abundance of Ochrobactrum, Rhodoplanes, and Nitrospira. It was concluded that the treatments analyzed in this study induced modulations in the ileal microbiota of the chickens examined.

## 1. Introduction

The functionality of the intestine depends on the intestinal microbiome, which is essential for the metabolism and nutrient absorption in chickens. A stable microbiota exhibits high diversity of microbial genera in a delicate balance, enhancing the metabolic capacity of the intestine [1]. The population dynamics of the intestinal microbiome are influenced by age, nutrition, stress, infectious diseases, health status, probiotic consumption, and exposure to antibiotics [2].

Numerous nutrients have the capacity to positively or negatively modulate intestinal functionality, consequently affecting animal health and performance [3]. Among the nutrients that act as modulators of the microbiota, the immune system, and intestinal integrity, nutraceutical compounds stand out. These include prebiotics, probiotics, organic acids, symbiotics, exogenous enzymes, polyunsaturated fatty acids, and phytobiotics [4].

Antibiotics are synthetic drugs or derived from natural sources, used to destroy or inhibit the growth of microorganisms. However, they also play a beneficial role in the intestines of broiler chickens, influencing microbial colonization, mucosal gene expression, and immune development [4]. The mechanism of action of antibiotics can involve the destruction of the cell membrane and the reduction in metabolites such as ammonia. For chickens, the use of antibiotics as performance enhancers increases the availability of nutrients in the intestine, reduces amino acid catabolism and bile salt breakdown, thereby improving protein digestibility [5].

Probiotics improve host health by altering nutritional, immunological, and physiological parameters through the action of probiotic strains in balancing the microbiota [3]. The mechanism of action of probiotics is influenced by their location in the gastrointestinal tract, intestinal mucosal integrity, and transit time in the intestine. Probiotic preparations include Lactobacillus, Bifidobacterium, Enterococcus, Streptococcus, Bacillus, and Pediococcus [6]. Probiotics are essential for promoting weight gain, regulating feed intake, digestion, immune response, and reducing mortality rates in broiler chickens [7].

The primary impacts of probiotics on the chicken’s organism occur through the protection of beneficial intestinal microbiota against pathogenic microorganisms via competitive exclusion and antagonism, altering metabolism by enhancing digestive system activities, microbial activity, and reducing ammonia production, promoting immunity through the increased production of bacteriocins, macrophages, and cytokines, as well as controlling protozoa [2].

The microbial community in the gastrointestinal tract plays a crucial role as one of the primary determinants of accelerated growth in chickens [8,9], affecting their health and immunological status through its impact on nutrient digestion and absorption. Exploring the bacterial phylogeny of the chicken gastrointestinal tract is of great interest [10]. Understanding the composition of a healthy intestinal microbiota offers the opportunity to develop optimal strategies for microbiota modulation aimed at improving performance, immunity, and food safety in chicken meat production [11].

The analysis of intestinal microbiota diversity is conducted through next-generation sequencing. This methodology generates a large amount of data, allowing for the identification of bacterial genera that are not cultivable using classical microbiological techniques [12]. Next-generation sequencing for genus identification is performed through PCR amplification of the 16S rRNA gene [7].

Alpha diversity of a sample is determined by its richness and evenness, where the richness of an ecosystem is represented by the number of different species comprising a given sample (total number of OTUs), and evenness is a factor that estimates the similarity and differences between the relative abundances of the taxonomic groups present in the sample [13]. Higher levels of diversity favor the existence of organisms that are functionally redundant, whose predominance indicates greater stability of the intestinal microbiota [14]. Beta diversity, on the other hand, determines the number of taxa in different environments, where higher values of beta diversity are indicative of a low level of similarity, expressed in the form of distances [11,15].

Given the diversity of microorganisms with potential use as probiotics and their different modes of action, this study aimed to evaluate the effect of three different probiotics—Normal Avian Gut Flora (NAGF), multiple colonizing strain probiotics (MCSPs), and a non-colonizing single strain probiotic (NCSSP)—and enramycin as an antibiotic growth promoter (AGP) on the modulation of the ileal microbiota.

## 2. Materials and Methods

### 2.1. Study Location

The project was carried out in Goiânia, GO, Brazil (−16.67926° N–49.25629° W). The climate of the region is classified as tropical, with rainfall occurring from October to May and a dry season from June to September. The average temperature during this period ranges from 20 to 35 °C, and annual precipitation ranges from 1500 to 1800 mm. This study was duly approved by the Animal Use Ethics Committee (CEUA), under protocol number 084/20.

### 2.2. Experimental Design

A total of 1400 male Cobb^®^ broiler chicks, obtained from a commercial hatchery of a poultry company located in Itaberaí—GO, Brazil, were used. The one-day-old broiler chicks were housed in an industrial poultry house negative pressure type (dark house) with a capacity of 20,000 chickens, measuring 125 m (L) × 12 m (W) (1.500 m^2^), and oriented east–west. The poultry house had an artificial lighting system, using white LED lights and 25 lux (2.5 foot-candles) light intensity until the chicks were 7 days old, after which the light intensity was gradually decreased to 5–10 lux (0.5–1.0 foot-candles). An automated air conditioning system using nebulizers and an air intake with an evaporative plate, regulated based on recommendations for maximum and minimum comfortable temperature and humidity for the broilers. The ambient temperature ranged from 20–34 °C and relative humidity from 30–70%, depending on the age of the broilers and the COBB Broiler Management Guide.

The central area of the poultry house (basic masonry structure and with curtain on the walls and roof) was divided into 40 movable pens, each measuring 2.88 m^2^. Each pen housed 35 chicks were separated by catwalks, equipped with ten nipple drinkers and one tubular feeder from the beginning to the end of the experiment, with a stocking density of 12 chickens/m^2^. Rice straw was used as poultry litter, which was changed once.

The treatments were as follows: negative control (basal diet without antibiotic and without probiotic); positive control (basal diet with antibiotic); NAGF (basal diet with Normal Avian Gut Flora probiotics); MCSP (basal diet with multiple colonizing strain probiotics); and NCSSP (basal diet with non-colonizing single strain probiotics). The experimental design was completely randomized (CRD), with 5 treatments, 8 replicates, and 35 chickens per experimental unit, totaling 1400 chickens. The antibiotic and probiotics were administered orally during all experiment (1 to 42 days).

The antibiotic used as a growth promoter (AGP) in the positive control diet was enramycin at a dose of 5 ppm (pre-starter phase) and 10 ppm (starter and grower phases). The tested probiotics were included in the diets at all stages of the broilers’ rearing, added on top at the dosages recommended by the manufacturer (Table 1). Each probiotic has a basic composition of bacterial strains, which are detailed in Table 2.

All broiler feed was produced in the own feed factory, near the poultry house. The ingredients were weighed and mixed in a horizontal-type mixer (617 pounds—maximum capacity).

The experimental feeds were mash-based, consisting of corn and soybean meal. Ingredient composition was calculated using the tables from Rostagno et al. [16], and the nutritional levels were in accordance with the standards proposed by the integrating company (Table 3). The broiler feed was mash type and for the desired water consumption, and the water temperature was maintained at 10–14 °C (50–57 °F). Feeding and water consumption were provided ad libitum.

### 2.3. Microbial Genomic DNA Extraction and 16S rRNA Gene Sequencing

To verify the composition and diversity of the intestinal microbiota, 25 samples of ileal digesta were collected from 42-day-old chickens representing 5 experimental groups (A—positive control diet; B—negative control diet; C—NAGF; D—MCSP; and E—NCSSP).

The commercial kit “ZR Fecal DNA MiniPrep^®^” from Zymo Research was used to extract DNA from the samples, following the manufacturer’s recommended protocol. The extracted DNA was quantified by spectrophotometry at 260 nm. To assess the integrity of the extracted DNA, all samples were subjected to 1% agarose gel electrophoresis. A segment of approximately 460 base pairs from the hypervariable V3V4 region of the 16S rRNA gene was amplified using universal primers, as described in the methodology, with the following PCR conditions: 95 °C for 3 min, then 25 cycles of 95 °C for 30 s, 55 °C for 30 s, and 72 °C for 30 s, followed by a final step at 72 °C for 5 min.

From these amplicons, the metagenomic library was constructed using the commercial “Nextera DNA Library Preparation Kit” from Illumina^®^. The amplicons were pooled together and subsequently sequenced on the Illumina^®^ “MiSeq” sequencer [17].

### 2.4. Sequencing Data Analysis

The reads obtained from the sequencer were analyzed using the QIIME (Quantitative Insights into Microbial Ecology) platform [17], following a workflow that included the removal of low-quality sequences, filtering, chimera removal, and taxonomic classification. The sequences were classified into bacterial genera through the recognition of operational taxonomic units (OTUs), based on sequence homology when compared to a database. The sequences were compared using the 2017 update (SILVA V.128) of the SILVA ribosomal sequence database [18]. To generate the classification of bacterial communities through OTU identification, 15,900 reads per sample were used to normalize the data. This approach ensured that samples with different numbers of reads were not compared, thereby avoiding bias in the taxonomic analysis.

In alpha diversity metrics, the richness (number of taxonomic groups) of the microbial community was analyzed and Shannon’s index was used. Principal component analysis (PCA) based on Bray–Curtis was used to estimate the dissimilarity in the community structure (beta diversity). Linear discriminant analysis effect size (LEfSe) was performed to detect differentially abundant taxa across groups using the default parameters linear discriminant analysis (LDA > 2).

### 2.5. Statistical Analysis

The difference in observed alpha diversity among the groups was estimated using the Kruskal–Wallis’s test [19]. The effect of treatments on beta diversity metrics was assessed between groups using the PERMANOVA (Permutational Multivariate Analysis of Variance) test with multiple comparisons corrected by the Bonferroni test [19,20]. Differences in the relative abundances of taxonomic groups were estimated using ANOVA with post hoc tests, such as Tukey–Kramer, Games–Howell, or Tukey, as appropriate [21]. The mean Firmicutes/Bacteroidetes ratio was compared between groups using ANOVA with Tukey’s multiple comparison test (*p* < 0.05).

## 3. Results

The ileal digesta microbiome was evaluated for alpha and beta diversity, taxonomic composition, and differences in taxon abundances among the experimental groups: positive control (A), negative control (B), NAGF (C), MCSP (D), and NCSSP (E).

### 3.1. Ileal Microbiome

Of the 25 samples, 3 exhibited a low number of reads and could not be evaluated. The remaining 22 samples were subsampled to 22.000 reads and classified into 6.094 OTUs (Silva V.128). For future analyses among the groups, 533 OTUs were retained, each with a minimum sum of 10 reads. The groups were classified into seven phyla, with Firmicutes showing the highest prevalence across all groups (Figure 1).

The 533 OTUs were identified across 42 genera, of which 10 genera had an abundance greater than 0.1%. In all groups, the genus Lactobacillus was the most prevalent (Figure 2).

#### 3.1.1. Alpha Diversity

Alpha diversity was assessed and no significant differences in species richness was observed among the microbial profiles of the five groups. However, a significant difference in the Shannon index was found between group B (negative control) and group D (MCSP—basal diet with multiple colonizing strain probiotics) (*p* = 0.02) (Figure 3). For more details, refer to the table of all samples.

#### 3.1.2. Beta Diversity

No clustering was observed between the groups, nor was there any statistical differences. The overall mean Bray–Curtis index was 66% dissimilarity (min. 4%–max. 97%; SD—standard deviation: 0.2; Figure 4).

#### 3.1.3. LEfSe

Of the OTUs identified in the intestinal microbiota of animals in groups A (positive control) and B (negative control), four OTUs were statistically different between the two groups, each with a linear discriminant analysis (LDA) score above 2. The most significant OTUs in the microbiota of animals in group A were *Lactobacillus* (OTU352) and *Lactobacillus* (OTU4). In group B, the most significant OTUs were *Kaistobacter* (OTU342) and *Lactobacillus* (OTU93) (Figure 5).

Of the OTUs identified in the intestinal microbiota of animals in the group A (positive control) and the group C (NAGF—basal diet + Normal Avian Gut Flora), 30 OTUs were statistically different between the two groups and had an LDA score > 2. The most significant OTU in the microbiota of animals in the positive control group was *Moryella* (OTU307). In group C, the most significant OTUs (based on LEfSe analysis) were *Lactobacillus* (OTU278, OTU191, and OTU407) and *Facklamia* (OTU200) (Figure 6).

Of the OTUs identified in the intestinal microbiota of animals in the group B (negative control) and the group C (NAGF), 27 OTUs were statistically different between the two groups and had an LDA score > 2. The most significant OTUs in the microbiota of animals in the negative control group were representatives of the genus *Lactobacillus* (OTU47 and OTU504). In the NAGF group, the most significant OTUs were *Lactobacillus* (OTU191 and OTU407) and *Facklamia* (OTU200 and OTU310) (Figure 7).

Of the OTUs identified in the intestinal microbiota of animals in the group A (positive control) and the group D (MCSP—basal diet with multiple strain colonizing probiotic), nine OTUs were statistically different between the two groups and had an LDA score > 2. The most significant OTU in the microbiota of animals in the positive control group was *Lactobacillus* (OTU216). In the group D, the most significant OTUs were *Lactobacillus* and *Clostridiales** (OTU516, OTU377, OTU411, OTU47, OTU341, OTU87, OTU111, and OTU228*) (Figure 8).

Among the OTUs identified in the intestinal microbiota of animals from the B (negative control) and D groups, 22 OTUs were statistically different between the two groups and had an LDA score > 2. OTU was more significant in the microbiota of the animals from the group B. *Lactobacillus* (OTU223, OTU147, OTU493, and OTU389) and *Corynebacterium* (OTU6) were the most significant OTUs in the microbiota of animals from the group D (Figure 9).

Of the OTUs identified in the intestinal microbiota of animals from the A (positive control) and E (NCSSP—basal diet with non-colonizing single strain probiotic) groups, three OTUs showed statistically significant differences between the two groups, with an LDA score > 2. The most significant OTU in the microbiota of animals from the positive control group was *Lactobacillus* (OTU216). *Lactobacillus* (OTU93) and *Chloroplast* (OTU461) were the most significant OTUs in the microbiota of animals from group E (Figure 10).

In contrast, in the microbiota of animals from the B (negative control) and E (NCSSP—basal diet with non-colonizing single strain probiotic) groups, three OTUs were statistically different between the two groups and had an LDA score > 2. No OTU was more significant in the microbiota of the animals from the group B. *Chloroplast* (OTU461), *Achromobacter* (OTU288), and *Ochrobactrum* (OTU11) were the most significant OTUs in the microbiota of animals from the group E (Figure 11).

## 4. Discussion

The intestinal microbiome of chickens is highly diverse [22,23] and interacts with the host organism, contributing to various functions including metabolism and immune maturation [24]. The decision to analyze the ileal microbiome in this study was based on the significance of this intestinal segment in maintaining homeostasis in poultry. The ileum is crucial for the absorption of water and minerals, as well as the digestion of starch and fat in broiler chickens [25,26].

Although the ileum does not primarily absorb most nutrients, it plays a crucial role in the final metabolism of dietary amino acids remaining from the proximal intestinal segments. This process prevents their escape and fermentation in the ceca, which could lead to putrefaction and the formation of toxic end products [27]. In this context, a healthy population of amino acid-absorbing bacteria in the ileum would be beneficial [28], underscoring the importance of investigating the composition of the ileal microbiome.

The identification of beneficial bacteria and the modulation of their abundance in the intestine can lead to advancements in animal husbandry [29]. Consequently, probiotics have increasingly been used in poultry farming to replace antibiotics and maintain chicken health [30]. In the present study, the supplementation of probiotics (NAGF, MCSP, and NSSPC) in the diet was tested to evaluate the composition of the ileal microbiome of broiler chickens and their potential as substitutes for antibiotics commonly used in feed.

NAGF (Normal Avian Gut Flora) is composed of a set of anaerobic bacteria, lactic acid-producing bacteria, and mannanoligosaccharides, which colonize the small intestine to the ceca of broilers, contributing to the control of enterobacteria (*Salmonella* sp. and *E. coli*) and thus improving the performance of chickens [31]. MCSPs (multiple colonizing strains) consists of 21 strains of lactic acid bacteria (primarily *Enterococcus* and *Lactobacillus*), aimed at enhancing performance and controlling pathogenic bacteria by colonizing primarily the small intestine of chickens [9] NCSSP (a non-colonizing strain) contains only *Bacillus subtilis*, a transient bacterium that lacks the fimbriae essential for mucosal adhesion and does not colonize any intestinal segment. The Bacillus-based probiotic aims to control pathogenic bacteria [9], such as *Clostridium perfringens*, responsible for necrotic enteritis [32].

At the phylum level, seven were detected. However, we found that the abundance of Firmicutes prevailed among all groups (Figure 1). This result is expected since the microbiome composition analysis was performed on chickens at 42 days old, and the Firmicutes phylum can increase its abundance to 60–65% in the ileal microbiome throughout a chicken’s lifespan [33].

Firmicutes is associated with energy uptake efficiency in animals [34], and its abundance may indicate adequate gastrointestinal functionality and eubiosis conditions in the gastrointestinal tract [35]. Therefore, these data support that the abundance of this phylum in the ileal microbiome, with a higher diversity of OTUs among the groups of probiotic-treated broilers (Figure 6, Figure 7, Figure 8, Figure 9, Figure 10 and Figure 11), may suggest an improvement in ileum function in broiler chickens.

The phylum Actinobacteria was also among the most abundant across the groups, although its proportion was significantly lower than that of Firmicutes. In contrast, the phylum Proteobacteria showed greater abundance only in group C (treated with NAGF). However, Proteobacteria have a relative abundance that can increase or decrease due to intrinsic or extrinsic factors that are not yet well understood [33], suggesting the need for further research to understand the possible association between the action of the NAGF probiotic and the abundance of the Proteobacteria phylum.

The dynamics of the ileal microbial communities are represented in Figure 2 by the 10 most abundant genera (greater than 0.1%). This analysis clearly showed the dominance of *Lactobacillus* in the ileum of all groups (Figure 2), a trend previously reported by other studies [36,37,38]. The small intestine of chickens is typically dominated by this genus, which belongs to the phylum Firmicutes [39,40,41,42].

The high dominance of *Lactobacillus* observed in 42-day-old chickens in this study can be explained by the fact that this bacterial genus colonizes the ileum at a later stage, with Enterobacteriaceae and Enterococcaceae being the most abundant taxa in the early days of the broiler’s life [28].

In addition to Lactobacillus, other representatives of the Firmicutes phylum were identified in this study and are described as colonizers of the ileum and other segments of the small intestine: *Turicibacter*, *Candidatus Arthromitus*, *Streptococcus*, and *Enterococcus* [28,39]. *Candidatus Arthromitus* can become the most abundant OTU in the ileal mucus in the early stages of a chicken’s life, while Lactobacillaceae are the most abundant in the lumen. However, the high abundance of *Candidatus Arthromitus* is temporary, as Lactobacillaceae become the most abundant family in both mucus and lumen [28], supporting our findings of a higher abundance of *Lactobacillus* compared to *Candidatus Arthromitus*. The genus *Facklamia* does not have a strong correlation with the ileal microbiome but has been described as a component of animal feed microbiota and poultry litter [43].

Some of the most prevalent genera followed distinct patterns of relative abundance across the different treatment groups. The abundance of the genus *Enterococcus* observed in group C, supplemented with MCSP (Figure 2), is unexpected since this genus is described as one of the first to colonize the ileal microbiota after hatching and tends to be replaced by other taxa during microbiota maturation [28]. Many species of the genus *Enterococcus* produce lactic acid [44], and some can even be used as probiotics [45]. However, there are also pathogenic species that act as a natural reservoir for antibiotic resistance genes [46]. Therefore, further studies are needed to identify the *Enterococcus* species observed in this group and to determine if there is a correlation between the use of the MCSP probiotic and the increase in this bacterial genus.

The other modulations identified involve genera with a low proportion of sequences in the microbiota, making it difficult to interpret their role’s relevance in the ileal microbiota of the analyzed chickens (Figure 2). Although no significant differences were identified between the microbial profiles of the five groups in terms of species richness and microbiome diversity in the present study (Figure 3), it has been observed that dietary supplementation with additives has previously increased these two parameters in supplemented chickens compared to untreated broilers [47,48]. Additionally, seasonality can impact the uniformity of OTUs in the ileal microbiome but not species richness [33,49].

Species richness in the ileum is lower compared to other intestinal segments. The cecum, for example, harbors a more complex and richer microbiota due to a longer digesta transit time, which favors extensive fermentation activity [50]. The lack of significant difference in species diversity among the microbial profiles of the five groups in this study can also be attributed to the ileal communities being less diverse, with the taxonomic representation specified by relative abundance more likely to diverge among individual groups [37].

The significant difference (*p* = 0.02) in the Shannon index between group B and group D found in this study may indicate greater diversity in the ileal microbiome promoted by the action of the probiotic with multiple colonizing strains (group D). The effect of multi-strain probiotics on microbial diversity arises from their role as growth promoters for certain OTUs, which can also influence the synthesis and metabolism of nutrients [51,52].

As observed in Figure 4, there was no clustering among the groups. The overall mean Bray–Curtis index of 66% dissimilarity may reveal different patterns of ileal microbiome composition among the groups of broilers subjected to treatments with different types of tested probiotics. Our results are consistent with the literature, which indicates that dietary supplementation with additives alone may not be a significant factor for the entire dataset nor when comparing different diets within groups [48]. These findings are supported by the results of Novoa et al. [33], who observed changes in microbiome similarity in broilers according to age. These authors found that ileal microbial communities were closely clustered in broilers at 50 days old and exhibited the greatest distance within a group of broilers at 56 days, indicating that clustering between or within the same group is variable and does not always occur.

Among the OTUs identified in the ileal microbiota of animals in group A (basal diet + antibiotic), *Lactobacillus* (OTU352) and (OTU4) were the most significant (*p* < 0.05; LDA > 2), supporting the hypothesis that enramycin (the antibiotic used in group A) can increase the abundance of *Lactobacillus* in broiler chickens fed with a diet containing this antibiotic [53,54]. However, changes in the cecal microbiome of broiler chickens induced by enramycin should be monitored with caution, as this antibiotic can promote alterations in the abundance of bacterial hosts carrying antibiotic resistance genes (ARGs) [52].

Meanwhile, *Kaistobacter* (OTU342) and *Lactobacillus* (OTU93) were the most significant OTUs in the microbiome of animals in group B (*p* < 0.05; LDA > 2), as observed in Figure 5. *Kaistobacter* has been associated with the suppression of bacterial diseases in plants [55]. Although microbial communities play important roles in disease suppression and the performance of broiler chickens, little is known about this interaction.

According to LEfSe analysis, which is used to determine statistically different biomarkers between groups (Figure 6), *Lactobacillus* (OTU278, OTU191, and OTU407) and *Facklamia* (OTU200) were enriched in the ileal microbiome of broilers in group C (NAGF) compared to group A (positive control) (*p* < 0.05; LDA > 2). Additionally, *Lactobacillus* (OTU191 and OTU407) and *Facklamia* (OTU200 and OTU310) were also enriched OTUs in group C compared to group B (negative control) (*p* < 0.05; LDA > 2; Figure 7), suggesting these OTUs as potential biomarkers for broilers supplemented with the Normal Avian Gut Flora probiotic.

The possible enrichment of *Facklamia* in the microbiome of broilers in group C may suggest an important action of the NAGF probiotic, as this bacterium has been associated with peripheral immunological indicators and intestinal mucosal health in the ileal microbiota of broiler chickens [56].

Of the nine OTUs that were statistically different (*p* < 0.05) between groups A and D (MCSP—probiotic with multiple colonizing strains), *Lactobacillus* (OTU516, OTU377, OTU411, OTU47, OTU341, OTU87, and OTU111) and *Clostridiales* (OTU228) were enriched in the ileal microbiome of broilers treated with MCSP compared to those in group A (*p* < 0.05; LDA > 2; Figure 8). This result corroborates the findings of Novoa et al. [33], where *Clostridiales* and *Lactobacillus*, belonging to the phylum Firmicutes, were also prominent OTUs in the microbiome of broiler chickens. In groups B and D, 22 OTUs were found to be statistically different (*p* < 0.05; LDA > 2), but no OTU was more significant in the microbiome of group B. In contrast, *Lactobacillus* (OTU223, OTU147, OTU493, and OTU389) and *Corynebacterium* (OTU6) were enriched in the microbiome of group C compared to group B (*p* < 0.05; LDA >2; Figure 9).

As presented in the LEfSe analysis in Figure 10, compared to group A, *Lactobacillus* (OTU93) and *Chloroplast* (OTU461) were more abundant OTUs in group E (NCSSP—non-colonizing single-strain probiotic (*p* < 0.05; LDA > 2)). Additionally, the LEfSe analysis showed that the abundance of *Chloroplast* (OTU461), *Achromobacter* (OTU288), and *Ochrobactrum* (OTU11) was enriched in group E compared to group B (*p* < 0.05; LDA > 2; Figure 11). The abundance of the genus *Achromobacter* in broiler chickens has been positively correlated with fatty acid concentration in the gastrointestinal tract [57]. The abundance of *Ochrobactrum*, belonging to the phylum Proteobacteria, requires further investigation since it is classified as an opportunistic bacterium and has been associated with increased diversity in ducklings with necrotizing colitis [58].

Based on the LEfSe analysis, the results of our study suggest an impact of probiotic use (groups C, D, and E) on the abundance of Lactobacillales, a trend previously reported by other authors [33,59,60].

## 5. Conclusions

This study provides insights into the effects of three types of probiotics on the ileal microbiome of 42-day-old broiler chickens. Our results showed that the phylum Firmicutes prevailed in all groups in this study, with a notable presence of the genus *Lactobacillus*. A greater variety of *Lactobacillus* OTUs was found in the groups of broilers treated with probiotics, particularly the multiple colonizing strains probiotic. *Clostridiales* were also enriched in the ileal microbiome of broilers treated with the multiple colonizing strains probiotic, suggesting a more diverse action of this type of probiotic on the microbiome richness of the broilers. The enrichment of *Facklamia* in the ileal microbiome of broilers supplemented with the Normal Avian Gut Flora probiotic warrants further investigation, as it may serve as an immunological biomarker.

## Figures and Tables

**Figure 1 microorganisms-12-01639-f001:**
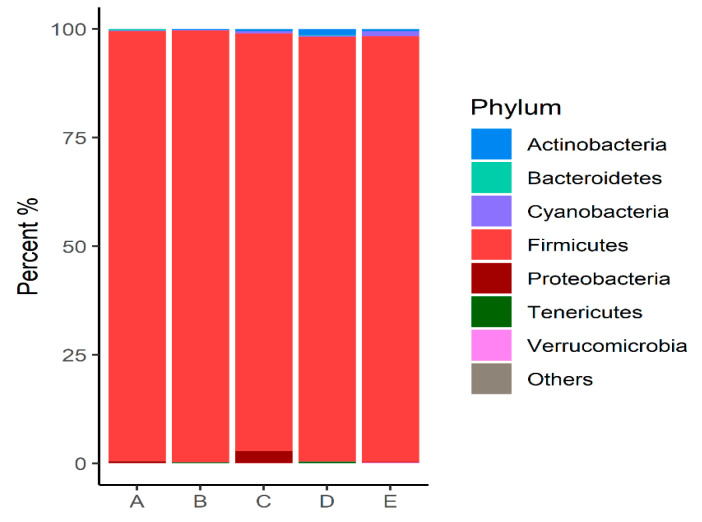
Relative abundance (>0.1%) of bacterial communities at the phylum level identified in the intestinal microbiota of animals subjected to different diets.

**Figure 2 microorganisms-12-01639-f002:**
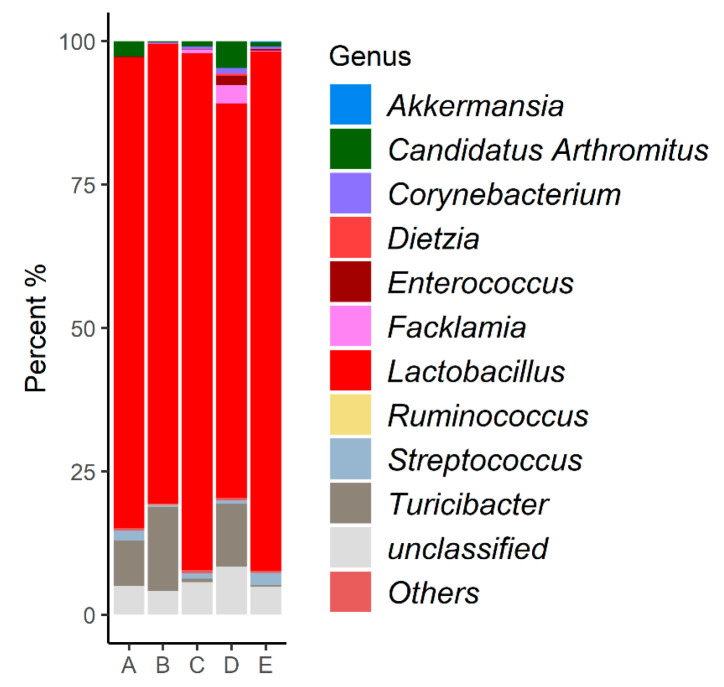
Relative abundance (>0.1%) of bacterial communities at the genus level identified in the intestinal microbiota of animals subjected to different diets.

**Figure 3 microorganisms-12-01639-f003:**
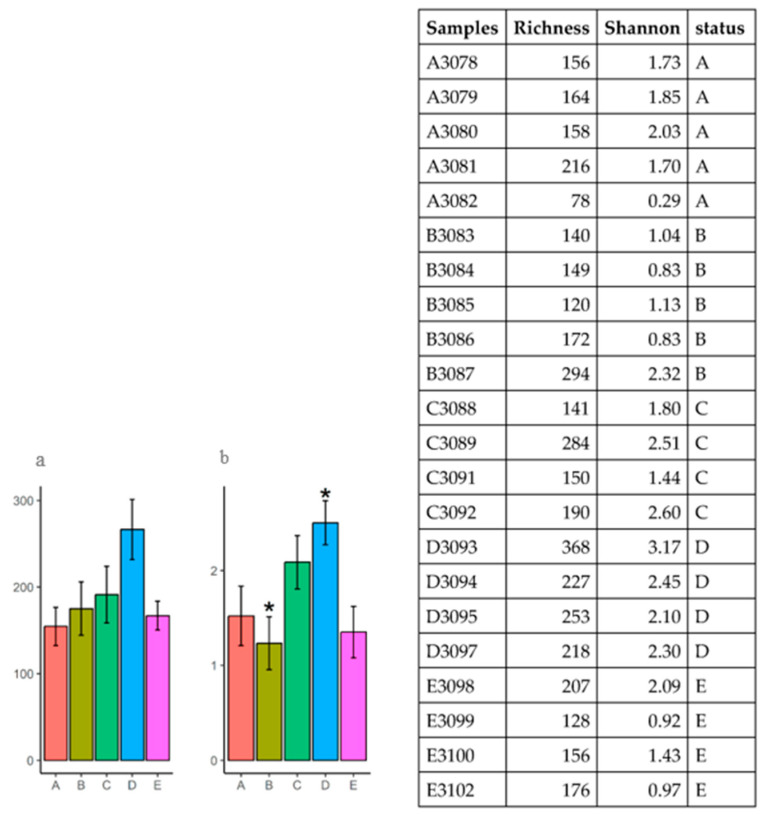
Diversity analysis of intestinal microbial profiles in broilers subjected to different diets. (**a**) Observed species richness or number of OTUs per sample; (**b**) Shannon diversity index. * Significant difference in the Shannon index (*p* = 0.02).

**Figure 4 microorganisms-12-01639-f004:**
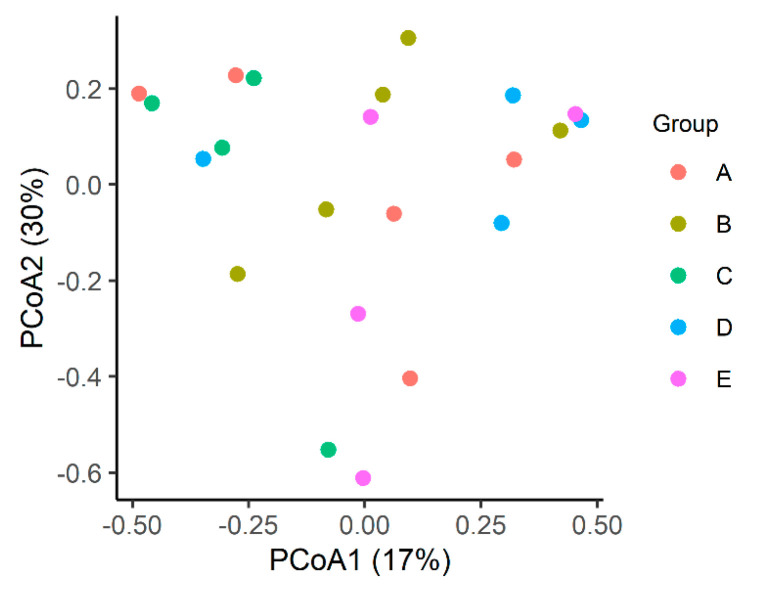
Two-dimensional ordination of intestinal microbial profiles in broilers subjected to different diets by principal coordinates analysis (PCoA).

**Figure 5 microorganisms-12-01639-f005:**
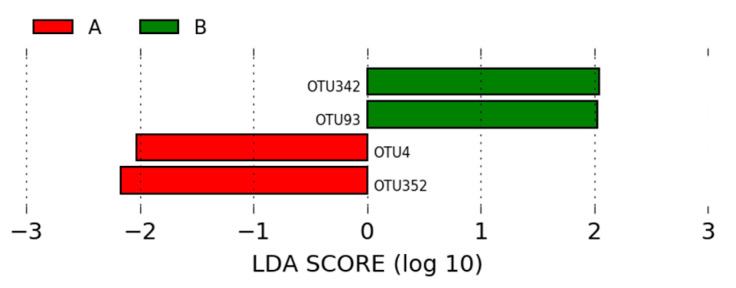
LEfSe analysis of OTUs showing phylum and the most specific classification of significant OTUs (LDA > 2) in groups A (positive control—basal diet with antibiotic) and B (negative control—basal diet without antibiotic and without probiotic).

**Figure 6 microorganisms-12-01639-f006:**
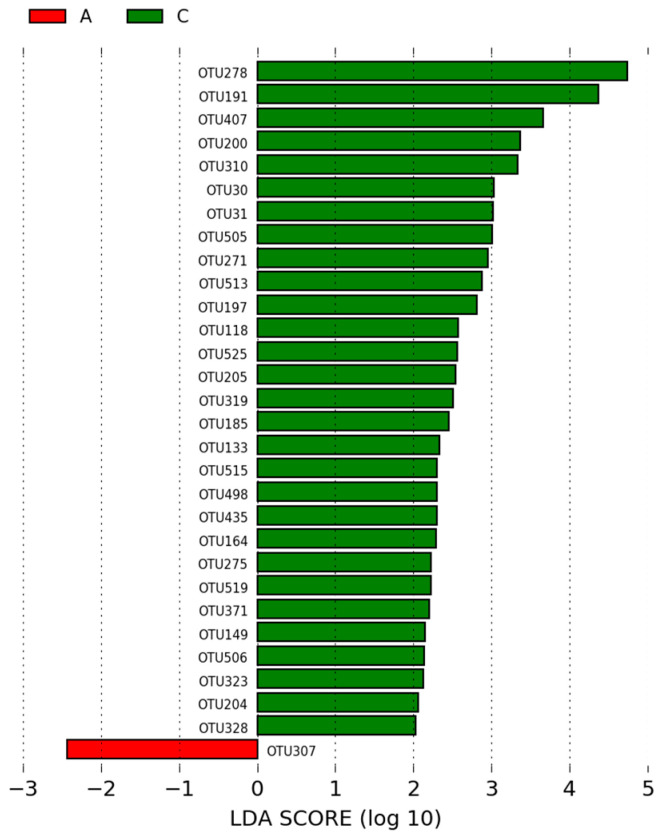
LEfSe analysis of OTUs showing phylum and the most specific classification of significant OTUs (LDA > 2) in groups A (positive control—basal diet with antibiotic) and C (NAGF—basal diet with Normal Avian Gut Flora probiotic).

**Figure 7 microorganisms-12-01639-f007:**
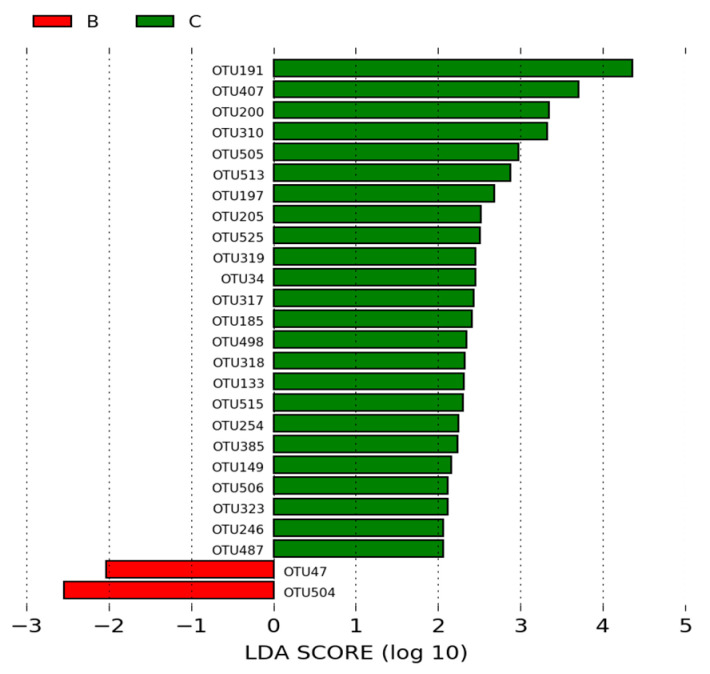
LEfSe analysis of OTUs showing phylum and the most specific classification of significant OTUs (LDA > 2) in groups B (negative control—basal diet without antibiotic and without probiotic) and C (NAGF—basal diet with Normal Avian Gut Flora).

**Figure 8 microorganisms-12-01639-f008:**
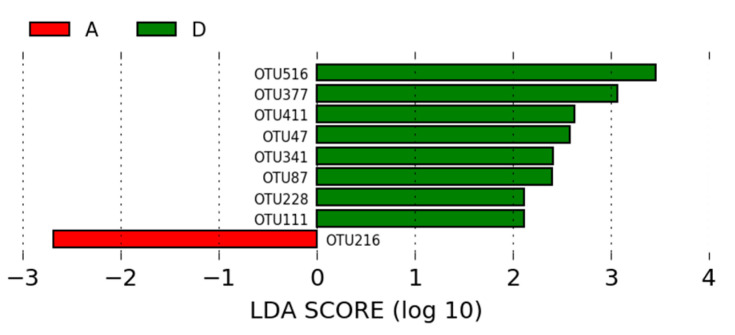
LEfSe analysis of the OTUs, showing phylum and the most specific classification of significant OTUs (LDA > 2) in groups A (basal diet with antibiotic) and D (MCSP—basal diet with multiple strain colonizing probiotic).

**Figure 9 microorganisms-12-01639-f009:**
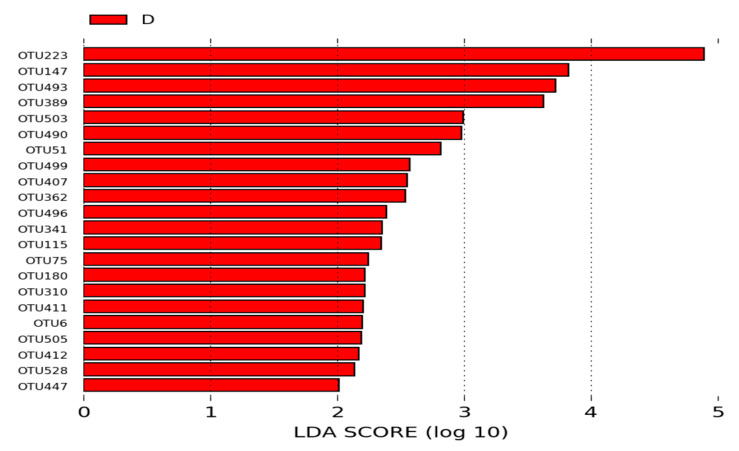
LEfSe analysis of OTUs showing phylum and the most specific classification of significant OTUs (LDA > 2) in groups B (negative control) and D (MCSP—basal diet with multiple colonizing strain probiotics).

**Figure 10 microorganisms-12-01639-f010:**
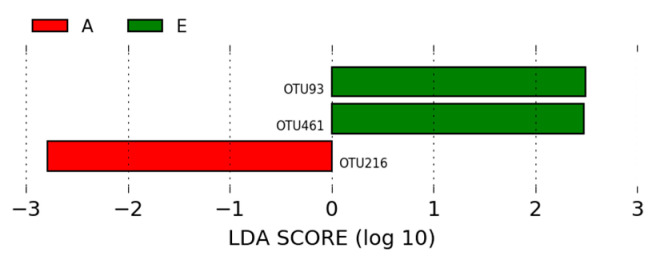
LEfSe analysis of the OTUs, showing phylum and the most specific classification of significant OTUs (LDA > 2) in groups A (positive control—basal diet with antibiotic) and E (NCSSP—basal diet with non-colonizing single strain probiotic).

**Figure 11 microorganisms-12-01639-f011:**
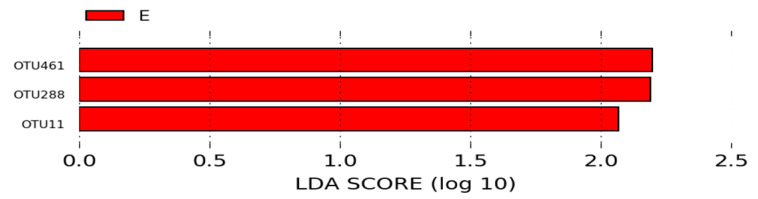
LEfSe analysis of the OTUs, showing phylum and the most specific classification of significant OTUs (LDA > 2) in groups B (negative control) and E (NCSSP—basal diet with non-colonizing single strain probiotic).

**Table 1 microorganisms-12-01639-t001:** Dosage recommendations for the probiotics used in the experiment.

Probiotics	Pre-Starter (g/t of Feed)	Starter (g/t of Feed)	Grower (g/t of Feed)	Finisher (g/t of Feed)
NAGF	500	100	100	100
MCSP	250	50	50	50
NCSSP	100	100	100	100

**Table 2 microorganisms-12-01639-t002:** Basic composition of the probiotics used.

Probiotics	Probiotic Strains—Composition
NAGF	Total anaerobic bacteria 1.0 × 10^4^; lactic acid-producing bacteria 1.0 × 10^4^; Mannanoligosaccharides 370 g/kg.
MCSP	*Bacillus subtilis* 1.0 × 10^8^; *Enterococcus faecium* 6.0 × 10^8^; *Lactobacillus acidophilus* 1.0 × 10^8^; *Lactobacillus delbrueckii* 1.0 × 10^8^; *Lactobacillus plantarum* 3.0 × 10^8^; *Lactobacillus reuteri* 5.0 × 10^8^; *Lactobacillus salivarius* 1.0 × 10^8^; *Pediococcus acidilactici* 3.0 × 10^8^.
NCSSP	*Bacillus subtilis* 1.0 × 10^8^

**Table 3 microorganisms-12-01639-t003:** Percentage composition and energy levels in the experimental diets by phase, prepared based on the nutrient requirement of broilers [16].

Ingredients (%)	Pre-Starter 1–7 Days Old		Starter 8–21 Days Old	Grower 22–35 Days Old	Finisher 36–42 Days Old
Corn	58.94		63.44	66.20	76.96
Soybean meal	33.93		28.80	26.33	13.87
Meat and bone meal	0.00		0.00	0.000	1.00
Limestone	1.24		1.09	1.23	0.86
Salt	0.31		0.31	0.34	0.28
Sodium bicarbonate	0.08		0.01	0.00	0.00
DL-Methionine	0.34		0.30	0.27	0.19
Poultry fat	0.53		0.60	1.93	0.53
Viscera meal	3.40		4.47	2.20	3.00
Feather and bloos meal	0.00		0.00	0.00	2.47
Acidifier	0.10		0.10	0.10	0.10
Antifungal	0.05		0.00	0.00	0.00
Monocalcium phosphate	0.43		0.21	0.68	0.00
Choline	0.02		0.02	0.03	0.02
L-lysine	0.39		0.37	0.40	0.52
L-Threonine	0.12		0.10	0.10	0.08
Avilamycin	0.005		0.01	0.01	0.00
Monensin + Nicarbazin	0.00		0.05	0.00	0.00
Antioxidant	0.004		0.004	0.004	0.004
Vitamin premix	0.05		0.05	0.05	0.05
Mineral premix	0.05		0.05	0.05	0.05
Monensin	0.00		0.00	0.03	0.00
Phytase	0.011		0.016	0.01	0.016
Total	100.00		100.00	100.00	100.00
Nutritional Levels	Pre-starter		Starter	Grower	Finisher
Metabolizable energy	Kcal	3.010	3.080	3.239	3.268
Crude protein	%	23.84	22.44	19.38	18.30
Total Lysine	%	1.60	1.50	1.30	1.21
Total methionine	%	0.74	0.68	0.56	0.49
Total methionine + cysteine	%	1.39	1.31	1.18	1.14
Total threonine	%	1.27	1.20	1.07	1.02
Total tryptophan	%	0.44	0.42	0.36	0.34
Digestible arginine	%	1.36	1.25	1.03	0.93
Digestible lysine	%	1.33	1.23	1.06	0.97
Digestible methionine	%	0.66	0.60	0.50	0.43
Digestible meth + cysteine	%	0.98	0.91	0.78	0.72
Digestible threonine	%	0.88	0.81	0.70	0.64
Digestible tryptophan	%	0.26	0.24	0.19	0.17
Calcium (Ca)	%	0.96	0.90	0.83	0.80
Total phosphorus (P)	%	0.68	0.64	0.58	0.54
Available phosphorus (P)	%	0.47	0.44	0.40	0.37
Sodium	%	0.22	0.20	0.19	0.19
Electrolyte balance	mEq/kg	246.57	214.5	169.72	147.77
Choline	ppm	2.000	1.850	1.650	1.550
Ca/P available	%	2.05	2.05	2.10	2.15

Vitamin–mineral supplements, guaranteed levels per kilogram of product: Folic acid, 1600.00 mg; Pantothenic acid, 24.96 g; Biotin, 80 mg; Butylated hydroxytoluene, 100 mg; Niacin, 67.20 g; Selenium, 600 mg; Vitamin A, 13,440,000 IU; Vitamin B1, 500 mg; Vitamin B12, 9200 mcg; Vitamin B2, 9600 mg; Vitamin B6, 4992 mg; Vitamin D3, 3,200,000 IU; Vitamin E, 21,000 IU; Vitamin K3, 2880 mg; Copper, 15 g; Iron, 90 g; Iodine, 1500 mg; Manganese, 150 g; Zinc, 140 g.

## Data Availability

The data presented in this study are available upon request from the corresponding author (due to privacy reasons).

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
