# Peer review of "Intestinal Microbiome Profiles in Broiler Chickens Raised with Different Probiotic Strains"

_microorganisms, 2024, doi:10.3390/microorganisms12081639_

Round 1

Reviewer 1 Report

Comments and Suggestions for Authors

The paper treats important aspect in the animals science and practice- the use of probiotics in the nutrition of animals and poultry. The latter are a suitable and preferable alternative to antibiotics.

The main aim of the present research is to to evaluate the effect of  three different probiotics and an antibiotic on the performance of broilers. The aim is very clearly defined and achieved by the authors.

The introduction contains the necessary information about the state of the art of the study and convinces in the necessity of doing this particular research. The material and method section is described in sufficient details, in my opinion even too many details about the birds housing. A minor remark is that the first line in Table 2 contains phrase in other language than English. In table 3 is this composition analysed by the authors or claimed by the producer?

The results are clearly presented. Large discussion is also provided in the manuscript.

The conclusions are supported by the data.

Author Response

Comments 1: A minor remark is that the first line in Table 2 contains phrase in other language than English.

Response 1: First line -Table 2: We translated it into English. Thank you for pointing this out.

Comments 2: In table 3 is this composition analysed by the authors or claimed by the producer?

Response 2: Thank you for pointing this out. We agree with this comment. The composition was prepared based on nutrient requirement of broilers. We added this information and the reference to the nutritional requirements manual in the caption of table 3.

Thank you for the corrections to improve the manuscript

Sincerely,

Authors

Reviewer 2 Report

Comments and Suggestions for Authors

This study aimed to evaluate the effect of three different probiotics—Normal Avian Gut Flora (NAGF), multiple colonizing strain probiotics (MCSP), and a non-colonizing single strain probiotic (NCSSP)—and an antibiotic on the modulation of the ileal microbiota. The results obtained are important for poultry practice, microbiologists, vaccine and probiotic manufacturers.

General comments:

In my opinion, the article should be supplemented with the following information:

In the Abstract section, there is no information on the number and sex of birds used in this study, the duration and method of antibiotic or probiotic administration, and the date of the ileal microbiota assessment.

In the Materials and methods chapter there is no information about

a. environmental parameters: temperature, relative humidity, light intensity and color, type of light (incandescent, fluorescent?)

b. type of building (no windows, no run)

c. What was the form of the feed mixture given to the birds?

d. method and date of administration of the vaccine and probiotics

Other: Please prepare the "Tables and "References" chapters in accordance with the requirements for authors.

Table headings must be in bold

In References chapter please use a "dot" after each abbreviation, for example J. Appl. Poult. Res. instead of J Appl Poult Res

Volume number in italic

Year in bold for all references

For significance, use a low letter "p" in italic instead of the  "p" in the main article

Specific comments

L6 on the performance of birds? If you have growth performance results (initial BW, final BW, BWG (g/d), FI, FCR, % martality), please add these results and a description in this article. This will significantly increase the value of this research to poultry practice.

L6, only the effect of antibiotics or probiotics on the composition of the ilial intestinal microbiota was studied

L7 please provide the name of the antibiotic and its type (AGP or therapeutic or prophylactic?)

L11 I suggest „ileal digesta” or „ileal microbiota”

L81 an enramycin antibiotic?

L97 Is the building closed without windows and catwalks?

Table 3 Provide the dates of use of pre-starter, starter, grower, finisher feed mixtures, and in line 123, the form (crumble, granulate) and name of the feed. Method of feeding ad libitum, temperature and water.

In Table 3, the sum of ingredients in the diets - pre-starter, starter, grower, finisher feed mixtures must be 100%. Currently there are 99,999 (pre-starter, 99,999 (Starter), 100,004 (grower), 99,998 (finisher)

In Table 3 for Choline „ppm” instead of Ppm

L136 ileal digesta or ileal microbiota

L259 "Significant" in the satistic sense?

L290 OTU87, OTU228, and .. instead of currebt form, please add OTU228

L322 and E (NCSSP-, instead of current form, add „E”

In References chapter please use a "dot" after each abbreviation, for example Front. Physiol. instead of Front Physiol

Volume number in italic

Year in bold for all references

 Full page range, for example 1835-1850 instead of 1835-50.

L549-550 Commun. Biol. 2022, 5, 293….instead of current form.

Author Response

Comments 1: In the Abstract section, there is no information on the number and sex of birds used in this study, the duration and method of antibiotic or probiotic administration, and the date of the ileal microbiota assessment.

Response 1: Thank you for pointing this out. We agree with this comment. Lines 6-7: We add the information about number and sex od birds (1.400 male Cobb® broiler raised up to 42 days of age). Lines 10-11: the duration and method of antibiotic or probiotic administration was described (The antibiotic and probiotics were administered orally during all experiment (42 days), mixed with broiler feed). Lines 12-13: The samples of ileal content were collected from 42-day-old chickens.

Comments 2: In the Materials and methods chapter there is no information about: a. environmental parameters: temperature, relative humidity, light intensity and color, type of light (incandescent, fluorescent?); b. type of building (no windows, no run); c. What was the form of the feed mixture given to the birds? d. method and date of administration of the vaccine and probiotics;

Response 2: Thank you for pointing this out. We agree with these comments.

a. Lines 95-103: We added environmental parameters: temperature and relative humidity (varied depending of chicken age – regulated based on recommendations for maximum and minimum comfortable temperature and relative humidity); poultry house had an artificial lighting system, using white LED lights and 25 lux (2,5 foot-candles) light intensity until 7 days of age, after which the light intensity was gradually decreased to 5-10 lux (0,5-1,0 foot-candles). The management and environmental parameters of the poultry house followed the instructions in the Cobb Broiler Management Guide.

b. Line 95: The type of building was an industrial poultry house negative pressure type (dark house).

c. Lines 122-124: The broilers feed were prepared in the university's own factory and feed were mixed in a horizontal type mixer (maximum capacity of 617 pounds).

d. Lines 115-116: The antibiotic and probiotics were administered orally (mixed in chicken feed) during all experiment (1 to 42 days). Lines 117-118: The antibiotic dosage was 5 ppm (pre-starter phase) and 10 ppm (starter and grower phases). Table 1 has the dosage for the probiotics used in the experiment.

Comments 3: Other - Please prepare the "Tables and "References" chapters in accordance with the requirements for authors. Table headings must be in bold. In References chapter please use a "dot" after each abbreviation, for example J. Appl. Poult. Res. instead of J Appl Poult Res. Volume number in italic. Year in bold for all references.

Response 3: Thank you for pointing this out. We agree with these comments. Now, all table headings are in bold and we corrected all references.

Comments 4: For significance, use a low letter "p" in italic instead of the "p" in the main article

Response 4: We agree with these comments and now all letter “p” (for significance) is in italic.

Comments 5: L6 on the performance of birds? If you have growth performance results initial BW, final BW, BWG (g/d), FI, FCR, % martality), please add these results and a description in this article. This will significantly increase the value of this research to poultry practice.

Response 5: Thank you for pointing this out. It was possibly an error in drafting the sentence. We adjusted the writing to better understand the text.

Comments 6: L6, only the effect of antibiotics or probiotics on the composition of the ilial intestinal microbiota was studied

Response 6: Thank you for pointing this out. Yes, we studied the effect of antibiotic (enramycin) and probiotics on the ileal microbiota. The decision to analyzed ileal microbiota was based on the importance of this intestinal segment in the final metabolism of dietary amino acids remaining from the proximal intestinal segments and other functions of the ileum such as absorption of water and minerals and homeostasis. Knowledge about the composition of the ileal microbiota can promote the healthy microbiota and contributes to amino acid-absorbing bacteria.

Comments 7: L7 please provide the name of the antibiotic and its type (AGP or therapeutic or prophylactic?)

Response 7: Thank you for pointing this out. We agree with these comments. The name and the type of antibiotic were provided on lines 10-11.  

Comments 8: L11 I suggest „ileal digesta” or „ileal microbiota”

Response 8: We agree with these comments. Line 13: We replace “ileal content” with “ileal digesta”

Comments 9: L81 an enramycin antibiotic?

Response 9: Thank you for pointing this out. Line 81-82: Yes, we insert the name of the antibiotic.

Comments 10: L97 Is the building closed without windows and catwalks?

Response 10: Yes. There are no windows in the poultry house. Catwalks existed inside the poultry house to separate the chicken pens from each treatment. The poultry house has the basic masonry structure and is sealed up with curtain on the walls and roof. We add this information in the lines 105-107.

Comments 11: Table 3 Provide the dates of use of pre-starter, starter, grower, finisher feed mixtures, and in line 123, the form (crumble, granulate) and name of the feed. Method of feeding ad libitum, temperature and water.

Response 11: We add this information about dates of use of pre-starter, starter, grower, finisher feed mixtures in Table 3. Lines 132-134: The chicken feed was mash type (produced in own feed factory) and for the desired water consumption, the water temperature was maintained at 10-14ºC (50-57ºF). Feeding and water consumption were ad libitum.

Comments 12: In Table 3, the sum of ingredients in the diets - pre-starter, starter, grower, finisher feed mixtures must be 100%. Currently there are 99,999 (pre-starter, 99,999 (Starter), 100,004 (grower), 99,998 (finisher)

Response 12: Thank you for pointing this out. Table 3: We check all quantities of diets ingredients and correct errors to obtain 100%.

Comments 13: In Table 3 for Choline „ppm” instead of Ppm

Response 13: We correct it, as suggested.

Comments 14: L136 ileal digesta or ileal microbiota

Response 14: Line 147 - We replace “ileal content” with “ileal digesta”

Comments 15: L259 "Significant" in the satistic sense?

Response 15: Thank you for pointing this out. Yes, in the statistic sense. Based on Lefse analysis (we added this information in the line 272)

Comments 16: L290 OTU87, OTU228, and .. instead of currebt form, please add OTU228

Response 16: We correct it, as suggested (Lines 300-301)

Comments 17: L322 and E (NCSSP-, instead of current form, add „E”

Response 17: We correct it, as suggested (Line 333).

Comments 18: In References chapter please use a "dot" after each abbreviation, for example Front. Physiol. instead of Front Physiol. Volume number in italic. Year in bold for all references.  Full page range, for example 1835-1850 instead of 1835-50. L549-550 Commun. Biol. 2022, 5, 293….instead of current form.

Response 18: Thank you for pointing this out. We agree with these comments. We corrected all references, as suggested (Lines 557- 698).

Thank you for the corrections to improve the manuscript

Sincerely,

Authors

Round 2

Reviewer 2 Report

Comments and Suggestions for Authors

This study aimed to evaluate the effect of three different probiotics—Normal Avian Gut Flora (NAGF), multiple colonizing strain probiotics (MCSP), and a non-colonizing single strain probiotic (NCSSP)—and an antibiotic on the modulation of the ileal microbiota . The results obtained are important for poultry practice, microbiologists, vaccine and probiotic manufacturers.

General comments:

The current version of the article takes into account all general comments from the previous review. However, it is necessary to slightly correct this article in terms of editorial requirements

Detailed comments:

Table headings must be in bold (see other articles in Microorganisms journal)

Table titles not in bold

L561 1835-1850 instead of 1835-50.

L658 303-351 instead of 303-51.

In the case of a journal without an abbreviated name, you do not put „a dot: after the journal name, for example Viruses 2022 instead of Viruses. 2022

Author Response

Comments 1: Table headings must be in bold (see other articles in Microorganisms journal). Table titles not in bold

Response 1: We apologize. Thank you so much for pointing this out. We had not understood the correction suggested in the first round of review. We saw at other articles in Microorganisms Journal as suggestion. We believe we have now made the correct change. The correction can be found in the lines 125 (Table 1. “Probiotics, pre-started g/t of feed, Started g/t of feed, Grower g/t of feed, Finisher g/t of feed”), 127 (Table 2. “Probiotics, Probiotics strains – Composition”) and 137 (Table 3. "Ingredients (%), pre-started 1-7-day-old, Started 8-21-day-old, Grower 22-35-day-old, Finisher 36-42-day-old”)

Comments 2: L561 1835-1850 instead of 1835-50.

Response 2: We agree with these comments. Corrections have been made and can be found in Line 561 (1835-1850).

Comments 3: L658 303-351 instead of 303-51.

Response 3: Thank you for pointing this out. We made the correction as suggested (Line 658: 303-351)

Comments 4: In the case of a journal without an abbreviated name, you do not put „a dot: after the journal name, for example Viruses 2022 instead of Viruses. 2022

Response 4: Thank you for pointing this out. We agree with these comments. We made all changes as suggested.

Thank you for the corrections to improve the manuscript

Sincerely,

Authors